# Feeding practices, dietary adequacy, and dietary diversities among caregivers with under-five children: A descriptive cross-section study in Dodoma region, Tanzania

**Walter C. Millanzi**[ORCID]*, Patricia Z. Herman, Bariki A. Ambrose

Department of Nursing Management and Education, The University of Dodoma, Dodoma, Tanzania

* walter.millanzi@udom.ac.tz, wcleo87@gmail.com

## Abstract

### Introduction

Suboptimal feeding practices among caregivers contribute to nutritional-related health problems in children in low and middle-income countries, such as Tanzania. The continuum of the recommended feeding practices has been an utmost challenge among caregivers during the transition from health facilities to homes for improved under-five children's health.

### Objective

This study assessed feeding practices, dietary diversities, and dietary adequacy among caregivers with under-five children in the Dodoma region, Tanzania.

### Methods

A community-based descriptive cross-sectional study in a quantitative research approach was adopted to study 289 randomly selected caregivers of under-five children in Dodoma region, Tanzania from March to August 2022. The World Health Organization Infant and Young Children Feeding Guidelines was the main data collection tool. Data were analyzed descriptively using a Statistical Package for Social Sciences computer software program version 25. Statistical limits were set at a 95% confidence interval and a 5% significance level.

### Results

Mean age was 26±6.47 of which 68.2% (n = 197) were females. Findings revealed 66.1% of caregivers had unsatisfactory feeding practices. It was observed that 67.8% of the caregivers fed their under-five children inadequately. Only 32.2% of them attained the minimum dietary diversity while 35.3% and 31.5% of caregivers demonstrated feeding practices < and > the recommended minimum Dietary diversity respectively. Moreover, 90.7% of caregivers fed their children group one food while 59.1% of them did not practice feeding the children

**Data Availability Statement:** All relevant data are within the paper and Supporting information files.

**Funding:** NO - This work did not receive any specific grant from funding agencies in the public, commercial, and or non-profit sectors. It was privately sponsored.

**Competing interests:** NO - The authors have declared that no competing interests exist.

snacks between meals. Feeding practice, dietary diversity, and dietary adequacy were significantly related to the caregiver's sex, education level, and socioeconomic status (p<0.05).

## Conclusion

Under-five children are at serious risk of nutritional problems as most caregivers in this study demonstrated unsatisfactory feeding practices on dietary diversities for dietary adequacy below the world health organization recommended Minimum Dietary diversity and Minimum Meal Frequency. Community-based nutritional education programs for caregivers need to be disseminated to the community level to address the problem accordingly.

## Introduction

Suboptimal feeding practices among caregivers contribute to nutritional-related health problems in children in low and middle-income countries [1]. However, with timely, appropriately adequate, and diversified feeding practices, under-five children's growth and development may prosper [2]. Existing knowledge uncovers that most caregivers may demonstrate satisfactory feeding practices when at health facilities due to the closest support, nutritional education, and guidance from healthcare professionals [3]. However, the continuum of appropriate feeding of under-five children within the recommended dietary adequacy and diversifications has been an utmost challenge among caregivers to maintain a smooth feeding transition from health facilities to home and improve children's health outcomes [4]. Unfortunately, it appears that comprehensive follow-ups and support at their homes are not widely available for many caregivers, particularly in the developing world [5].

This study assumes that cultural diversities of caregivers after discharge from health facilities have been sometimes associated with inappropriate and/or inconsistent feeding practices for their under-five children at home. The situation has also been linked with several adverse health outcomes that children face such as stunting, under-nutrition, over-nutrition, early hospital re-admissions, emergency room visits, longer hospital stays, and/or deaths [6, 7]. Global estimate highlights that approximately 232.9 million under-five children are malnourished with a proportional distribution of almost 149 million children stunted; 45 million are wasted and approximately 38.9 million children are overweight or obese [8]. The burden of nutritional-related health outcomes is substantially higher in Sub-Saharan African regions whereby an estimated 155 million under-five years children are stunted and 52 million wasted.

World Health Organization recommends newborns be exclusively breastfed for the first six months of life, then begin nutritionally adequate, safe, and appropriately fed complementary foods from six to 24 months to meet the evolving needs of the growing children [9]. It appears that despite the World Health Organization, recommended infant and young children feeding, the problem of over or under-feeding practices persists among caregivers [10]. Statistics of feeding practices, dietary diversity, and dietary adequacy indicate that only 41% of children are exclusively breastfed for six months, 25% receive the Minimum Dietary Diversity (MDD) and 51% receive the Minimum Required Meal Frequency (MMF) respectively [11]. The highest burden of inappropriate complementary feeding practices is in low and middle-income countries [12].

In developed countries such as the United Kingdom, Canada, Australia, the United States of America, and New Zealand just to mention a few, caregivers can practice appropriate feeding to their children because they are periodically visited at their homes or contacted via

telephone calls by nurses, midwives, and peers for continuum education, support, and guidance [13, 14]. Supportive home visiting and/or community-based clinics, while recognized for achieving positive attitudes and high satisfaction with feeding practices among caregivers and promoting health outcomes to children, the visits or telephone calls appear to be costly and not always universally available and accessible in the developing world Tanzania inclusive [9].

From the nutritional-related health conditions aforementioned earlier, there seems that the burden of under-five morbidity and mortality rates become significantly high, particularly in the developing world including Tanzania [15]. Tanzania is inclusive with the problem of nutrition for under-five year children of which Tanzania Demographic and Health Survey data shows that 59% of infants are exclusively breastfed for six months, 39% of 6–24 months children are given minimum recommended meal frequency and 26 percent are given minimum recommended diverse diet [16]. Despite the huge progress achieved in the country yet, the majority of children are not fed at the Minimum Dietary Diversity (MDD) [17]. The most common food types caregivers feed children include 91% grains, roots, and tubers and 65% Vitamin A containing fruits and vegetables [18]. Other foods include eggs (7%), meat and fish (36%), milk and dairy products (22%), legumes and nuts (35%), and other vegetables 21% [18, 19].

The aforementioned nutritional trend has resulted in the prevalence of 31% stunting, 6% wasting, and 14% underweight. Considering such a situation, it becomes unclear whether caregivers with under-five children are competent in practicing feeding, and dietary diversity to promote dietary adequacy at their homes. Available scholarly works [20–22] have demonstrated evidence on the domain of factors influencing feeding practices, the prevalence of nutrition disorders, and some interventional research that focused on empowering breastfeeding mothers and/or caregivers for appropriate feeding practices. However, there is limited information in the Dodoma region, Tanzania on the caregivers' competencies in demonstrating feeding practices, adequacy, and diversity to their under-five children at their homes.

Additionally, it remains unclear whether caregivers with under-five children at home continue to attend community-based clinics to receive feeding practice education and associated services to keep improving feeding practices for the positive health outcomes of their children. Thus, the study at hand intended to fill the gap by employing a descriptive cross-sectional study to quantify feeding practices, dietary adequacy, and dietary diversities among caregivers with under-five children in the region.

## Methods and materials

### Study setting

The study was conducted from March to August 2022 in Dodoma region, a central part of Tanzania. Being a newly growing area, political, academic, and business hub, the region is keeping in densely populated due to immigration leading to a high population of 3,085,625 [23]. The region features a semi-arid climate with warm to hot temperatures and little rainfall throughout the year, which is closely linked with challenges experienced by people there, not only in cultivating food and cash crops but also in keeping domestic animals. Thus, food security from plantations and dairy products become to be not easily available and accessible for the good health of children and adults. People in the region believe in and depend on natural foods such as mushrooms, insects, and vegetables because they may sometimes be easily available and accessible within their reach. The socio-cultural habits of people there are linked with low and inappropriate dietary intake for both children and adults. The region was therefore sampled owing to the growing prevalence of pediatric nutritional-related health outcomes

including the prevalence of stunting by 37.2% (HAZ <-2), moderate stunting by 26.1% (HAZ <-2 and > = -3), and severe stunting by 11.1% (HAZ>-3) [6].

## Study design

A community-based descriptive cross-sectional design in a quantitative approach was employed. This study was conducted based on the institutional guidelines alongside their research ethics and standards for undergraduate and postgraduate programs that provide a guiding framework for conducting research by adhering to ethical issues to meet national and international research standards [24].

## Study population and sample size determination

Based on the recommendations from previous scholars [25–27], The following procedures were performed to determine the minimum sample size for the study using the formula by Cochran 1977 [28].

$$n = \frac{Z^2 p(1-p)}{e^2} \tag{1}$$

Whereas; n = a minimum sample size of the study
p = the population proportion from previous studies (75.4%) [11]
$z^2$ = z-value (1.96) at reliability level (95%) or significance level (5%)
$e^2$ = acceptable sampling error (e = 0.05)

$$n = \frac{1.96^2\,0.754(1-0.754)}{0.05^2} \quad n = \frac{3.8416\,x\,0.754(0.246)}{0.0025} \quad n = \frac{0.71255533}{0.0025} = 285$$

Given the non-response rate of 10% (n = 29), the minimum sample size of the study was n = 314 participants. However, as shown in Fig 1, the study analyzed 289 of them because 1143

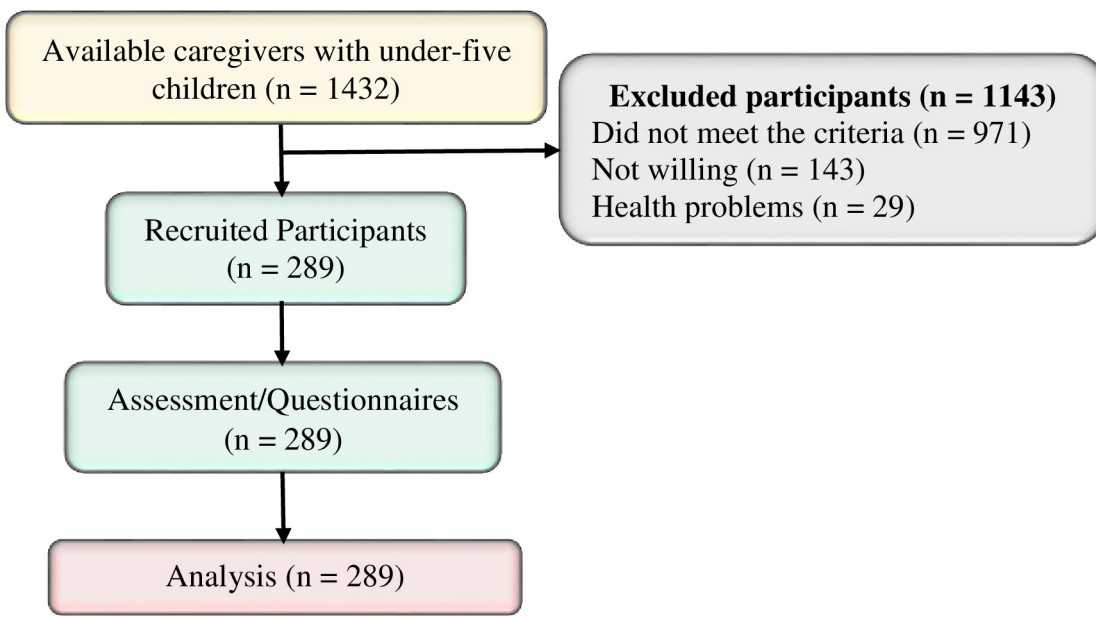

**Fig 1. A flow pattern of recruitment and sampling procedures of the study population. Source**: Study plan (2022).

out of 1432 available caregivers in the study setting were not studied and analyzed due to various reasons including not meeting the inclusion criteria (n = 971), not providing informed consent (n = 143) and reported health problems (n = 29).

## Sampling procedures

The region was sampled purposely due to the growing prevalence of pediatric nutritional-related health outcomes including the prevalence of stunting by 37.2% (HAZ <-2), moderate stunting by 26.1% (HAZ <-2 and > = -3), and severe stunting by 11.1% (HAZ>-3). Caregivers were reached in the community after being identified from a register available in the respective health facilities. Health facilities were used as media to access the permanent addresses of caregivers with under-five children who were then contacted and followed in the respective ward. Using a systematic random sampling technique via a random table number a required minimum sample was reached. A list of caregivers with under-five children was established from the daily attendance sheets of the respective health facilities where their addresses were obtained (n = 1432). Since, 1432/289 = 5, a 1-in-5 systematic sampling was performed. A random sampling started at point 3 and using random number tables the procedure continued from that on until a minimum required sample of 289 participants was reached.

## Inclusion and exclusion criteria

**Inclusion criteria.** caregivers who are biological parents or closest relatives with under-five children based on the predefined criteria including the provision of informed consent, a resident of Dodoma region, could communicate via Swahili language, breastfeeding, and those who started complementary feeding practices.

**Exclusion criteria.** Self-reported health problems that would affect participation and communication during the data collection process and those who reported being part of other projects.

## Data collection procedure

The principal investigator assisted by the trained research assistants collected data using interviewer-administered structured questionnaires to assess caregivers' feeding practices. Given the administrative support from a ward executive officer and village/street leaders, caregivers were reached at their homes following the addresses obtained from the health facility and gathered in a common room of one of the schools available in the study setting as negotiated by the school head of the respective school. Data collection procedures were performed by the principal investigator and supported by trained assistant researchers. Independent seats were used per study respondent to prevent copying and pasting or sharing colleagues' responses. The research team provided brief instructions to the respondents on how to fill out the questionnaires. The principal investigator was the first to read the items and let the respondents fill afterward. The team was available throughout the process to supervise, and respond to queries from the respondents before collecting and securing the filled-up questionnaires. Codes were used instead of participants' names in the questionnaires to assure confidentiality. Thirty (30) to forty-five (45) minutes were an approximate time for the completion of filling up the questionnaires.

## Data collection tools

The interviewer-administered infant and young children feeding (IYCF) checklist adopted from the World Health Organization (WHO) was the main data collection tool. The WHO's

IYCF checklist was structured and assessed feeding practice, dietary diversity, and adequacy. It included items that assessed group 1 foods (grains, roots, and tubers); group 2 foods (legumes and nuts); group 3 foods (Flesh foods, meat, fish, poultry, and liver/organ meats); group 4 foods (Eggs); group 5 foods (Vitamin A rich fruits and vegetables); group 6 foods (Dairy products: milk, yogurt, cheese); and group 7 foods (Other fruits and vegetables). Reproductive Health card 1 (RCH1) was the second data collection tool to confirm children's sociodemographic characteristics profiles and check the plotted weight and define it against the World Health Organization Z-scores to detect and conclude normality, underweight, or overweight. Children beyond 2 years of age were measured actual weight by using a weighing balance and their weights and heights were recorded accordingly. The tool was proceeded by sociodemographic characteristics profiles of caregivers and children respectively.

## Validity and reliability

**Validity.**    Content validity has been opted and it was assured in this study by developing items relevant and appropriate research tools, which were then shared with statisticians and expert colleagues for inputs on content appropriateness, sentence structure, language, and organization. While other aspects such as content appropriateness, sentence structure, and organization of the items remained unchanged, their responses required research tools to be translated into the Swahili language to blend with the literacy level of the study respondents and improve the clarity, understanding, accuracy, and completeness of the information. The tools were re-shared with them for their final proof with which there were no further inputs to be amended.

**Reliability.**    The principal investigator conducted the pre-test of the research tools to a sample of 10% (n = 30) respondents in an independent geographical location from the sampled study settings to prevent information contamination. Indicators such as the relevance of the items, language appropriateness, clarity, and duration it would take to finish filling the questionnaires. The inter-observer rating was employed to rate the relevance of the items among 10 consulted independent reviewers. Observation from a pre-test revealed that all items were relevant with a score range of 9/10 to 10/10, the language was appropriate and clear and the questionnaires would be filled and completed within a range of 30 to 60 minutes. Findings of the pre-test were then subjected to a scale analysis to determine the reliability measure of the tools which revealed that items that assessed feeding practice demonstrated a Cronbach $\alpha = 0.76$, while 0.81 for feeding adequacy scale, and 0.79 for the feeding diversity and thus, as recommended by previous scholars [29–31] the tools were considered reliable for the actual field data collection.

## Variable measurement

**Perceived feeding practices.**    The ten (10) items with 4 answer choices measured caregivers' perceived feeding practices of which each item carried a least of three (3) points and a maximum of four (4) points (Points ranged between 30 and 40). The highest points were considered positive perceived feeding practices otherwise not.

**Dietary diversity.**    This variable was measured based on the items prescribed in the WHO's IYCF checklist that categorized foods into seven groups. The groups included Group one (Grains, roots, and tubers); Group two (Legumes and nuts); Group three (Flesh foods: meat, fish, poultry, and liver/organ meats); Group four (Eggs); Group five (Vitamin A rich fruits and vegetables); Group six (Dairy products: milk, yogurt, cheese); and Group seven (Other fruits and vegetables). If a child consumed at least one food item from a food group on the same or the previous day, the group was assigned a value of one (1) for that child, and zero (0) if not consumed. The group points were summed up to obtain the dietary diversity score,

which ranged from zero to seven, whereby zero was considered as a non-consumption of any of the food items in the food groups, and seven represented the highest level of diet diversification. The MDD was attained if a child had consumed four or more food groups (FG $\geq$ 4) out of the seven food groups over the previous day.

**Dietary adequacy.** the variable was measured based on the Minimum Meal Frequency (MMF) criteria on the same or previous day including a snack. Criteria number one defines a minimum number of times that a child aged between 6 and 23 months received solid, semi-solid, or soft food; Criteria number two describes two times for breasting infants aged between 6 and 8 months; Criteria number three for three times of breastfeeding children aged between 9 and 23 months; and criteria number four for four times for non-breastfeeding children aged between 6 and 23 months. If the frequency a baby received solid, semi-solid, or soft foods was less than the aforementioned recommendations, then it was assigned as dietary inadequacy.

## Data analysis

Data were analyzed using a Statistical Package for Social Sciences computer software program version 25. Descriptive analysis established frequencies and percentages of caregivers and children's sociodemographic characteristics profiles. Nevertheless, Chi-square and cross-tabulation tests determined the prevalence and relationship between variables. Children's weights were crossed against WHO weight for age Z-score to determine underweight, overweight, or normal weight based on sex. Statistical limits were set at a 95% confidence interval and a 5% significance level.

## Results

### Sociodemographic characteristics profiles of caregivers

A sample of 289 caregivers with under-five children participated in this study with a response rate of 100%. Findings in Table 1 indicated that the mean age was 26 years±6.5, with a minimum age of 16 years and a maximum of 42 years while the most prominent age group (36.7%) ranged between 22 years and 27 years. Moreover, findings show that 68.2% (n = 197) of caregivers were females and 74.4% (n = 215) of them were married. Caregivers with secondary education accounted for 65.7% (n = 190) while those with more than five children accounted for 10% (n = 29). Findings indicate that 81.7% (n = 236) of caregivers delivered their children at health facilities and 57.1% (n = 165) of them were the proximal individual taking care of their under-five children. Other findings are shown in the Table 1.

### Sociodemographic characteristics profiles of caregivers' children

Findings presented in Table 2 show sociodemographic characteristics profiles of caregivers' children. The mean age of children was 2 yeras±1.2 while 58.5% (n = 169) of them were girls. The majority of children (34.9%) and (31.1%) were aged between 1 and 2 years respectively. 79.9% (n = 231) of them were vaccinated with all appropriate vaccines based on their age. Despite 54.0% (n = 156) of children having normal body weight, but, 13.8% (n = 40) of them were severely underweight, 18.3% (n = 53) moderately underweight, and 13.8% (n = 40) overweight respectively.

### Proportional distributions of feeding practice among caregivers with under-five children

Findings about feeding practices presented in Fig 2 show that 66.1% (n = 191) of caregivers had unsatisfactory dietary feeding practices for their children. The prevalence of good dietary

**Table 1. Sociodemographic characteristics profiles of caregivers (n = 289).**

| Variable | Frequency (%) |
|---|---|
| **Age: Mean age = 26 years±6.5** | |
| 16–21 years | 72(24.9%) |
| 22–27 years | 106(36.7%) |
| 28–33 years | 61(21.1%) |
| 34–39 years | 42(14.5%) |
| 40–42 years | 8(2.8%) |
| **Sex** | |
| Male | 92(31.8%) |
| Female | 197(68.2%) |
| **Education Level** | |
| No formal education | 15(5.2%) |
| Primary education | 19(6.6%) |
| Secondary education | 190(65.7%) |
| Tertiary education | 65(22.5%) |
| **Marital Status** | |
| Single | 54(18.7%) |
| Widow | 20(6.9%) |
| Married | 215(74.4%) |
| **Number of children** | |
| Beyond five children | 29(10.0%) |
| One child | 162(56.1%) |
| Between two and five children | 98(33.9%) |
| **Place baby was delivered** | |
| Home | 31(10.7%) |
| Traditional midwives | 22(7.6%) |
| Hospital | 236(81.7%) |
| **Average birth space** | |
| Less than one year average | 55(19.0%) |
| One year | 165(57.1%) |
| Two years average | 69(23.9%) |
| **Number of under-five children** | |
| Two under-five children | 72(24.9%) |
| One under-five child | 217(75.1%) |
| **Occupation** | |
| Don't have a job | 65(22.5%) |
| Employed | 92(31.8%) |
| Self-employed | 132(45.7%) |
| **Proximal childcare person** | |
| Helped by house girl/boy | 58(20.1%) |
| Helped by grandparents/ relatives | 66(22.8%) |
| Parent Him/ Herself | 165(57.1%) |
| **Residency** | |
| Rural | 54(18.7%) |
| Urban | 235(81.3%) |
| **Religion** | |
| Christian | 131(45.3%) |
| Muslim | 147(50.9%) |
| Pagan | 11(3.8%) |

**Source**: Field Data (2022).

**Table 2. Socio-demographic characteristics profiles of caregivers' children (n = 289).**

| Variable | Frequency (%) |
|---|---|
| **Children sex** | |
| Boy | 120(41.5%) |
| Girl | 169(58.5%) |
| **Children age distribution: Mean age = 2 years±1.2** | |
| 1 yrs. | 101(34.9%) |
| 2 yrs. | 90(31.1%) |
| 3 yrs. | 52(18.0%) |
| 4 yrs. | 25(8.7%) |
| 5 yrs. | 21(7.3%) |
| **Children's birth weight** | |
| Below 3.5 kgs. | 29(10.0%) |
| Above 4.4 kgs. | 18(6.2%) |
| Normal; between 3.5 and 4.4 kgs. | 220(76.1%) |
| Don't remember | 22(7.6%) |
| **Children vaccination status** | |
| Didn't get all appropriate vaccines/per age | 58(20.1%) |
| Got all appropriate vaccines/per age | 231(79.9%) |
| **Children's nutritional status; WHO w/a Z score** | |
| Severely Underweight | 40(13.8%) |
| Moderately Underweight | 53(18.3%) |
| Normal Weight | 156(54.0%) |
| Overweight | 40(13.8%) |

**Source**: Field Data (2022).

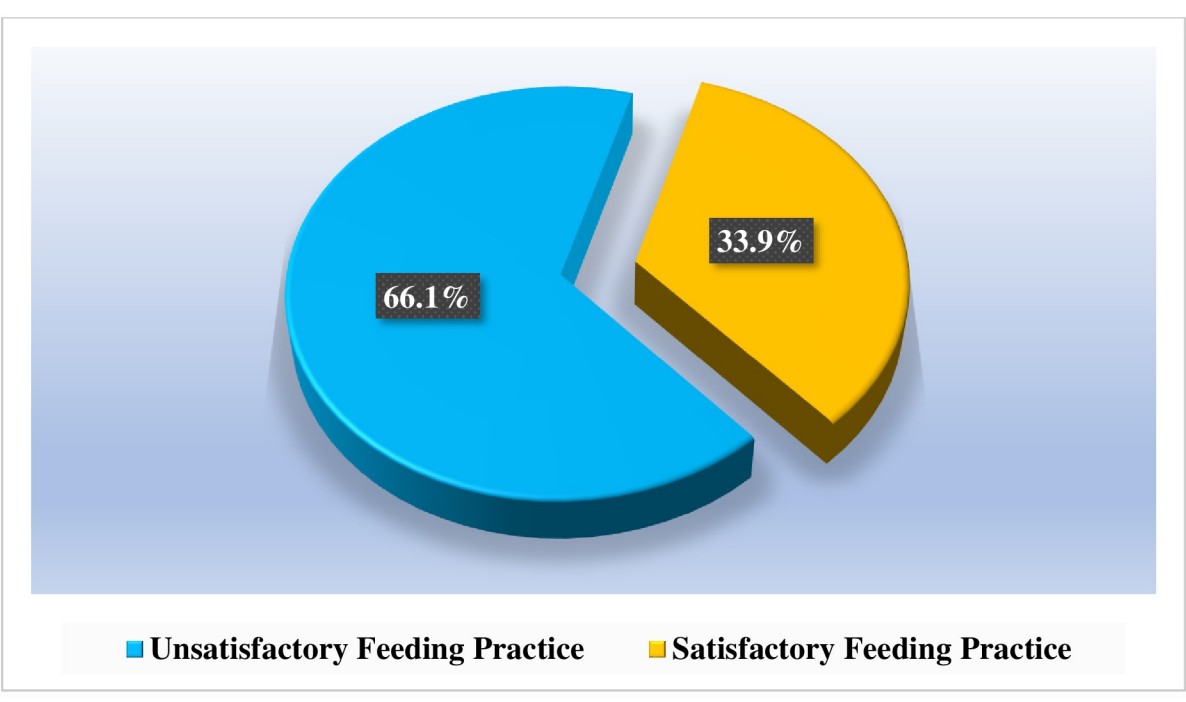

**Fig 2. Proportional distributions of feeding practices among caregivers with under-five children. Source**: Field Data (2022).

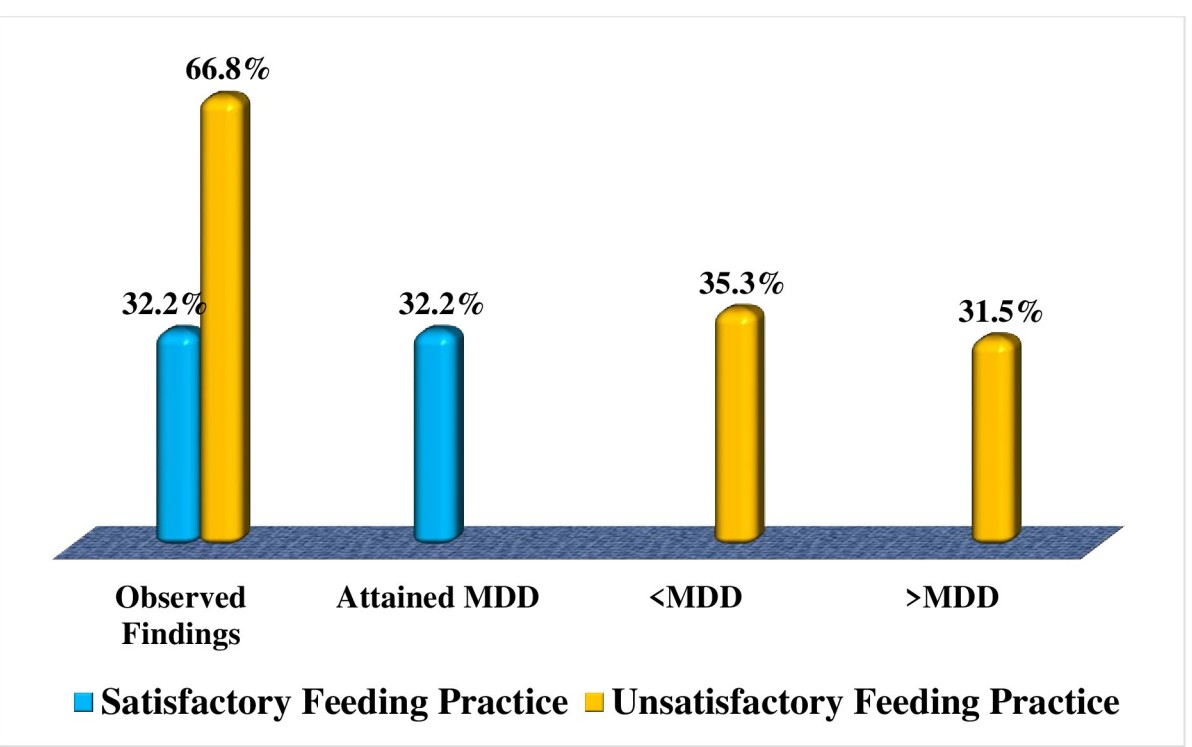

**Fig 3. Proportions distributions of caregivers' feeding practices of dietary diversity based on recommended minimum dietary diversity (MDD). Source**: Field Data (2022).

practice was therefore demonstrated by 33.9% (n = 98) of them. Refer to Fig 2 for the figurative elaboration of the findings.

### Proportions distributions of caregivers' dietary diversity feeding practices to their under-five children based on the recommended minimum dietary diversity (MDD)

As shown by the findings presented in Fig 3, it was observed that 66.8% (n = 202) of the caregivers practiced unsatisfactory dietary diversity feeding to their under-five children. The pattern of dietary diversification practices among caregivers indicated that 31.5% (n = 91) of caregivers fed their under-five children diverse foods beyond the recommended MDD whereas, 35.3% (n = 102) of them fed their under-five children foods less than the recommended MDD, and 33.2% (n = 96) of the caregivers attained the recommended MDD feeding. Refer to Fig 3 for the figurative presentation of the findings.

### Proportional distributions of caregivers' feeding practices of dietary diversity to their under-five children based on groups of foods

Findings of caregivers' proportional distributions of practicing dietary diversity when feeding their children are shown in Fig 4. It was found that 100% of caregivers practiced dietary diversity feeding. However, group one foods were the most option for the majority of caregivers (90.7%) when feeding their children followed by group two foods (88.6%), and group seven food (75.8%). Group five foods were the least (21.5%) to be fed to under-five children. Refer to Fig 4 for the figurative presentation of the observed findings.

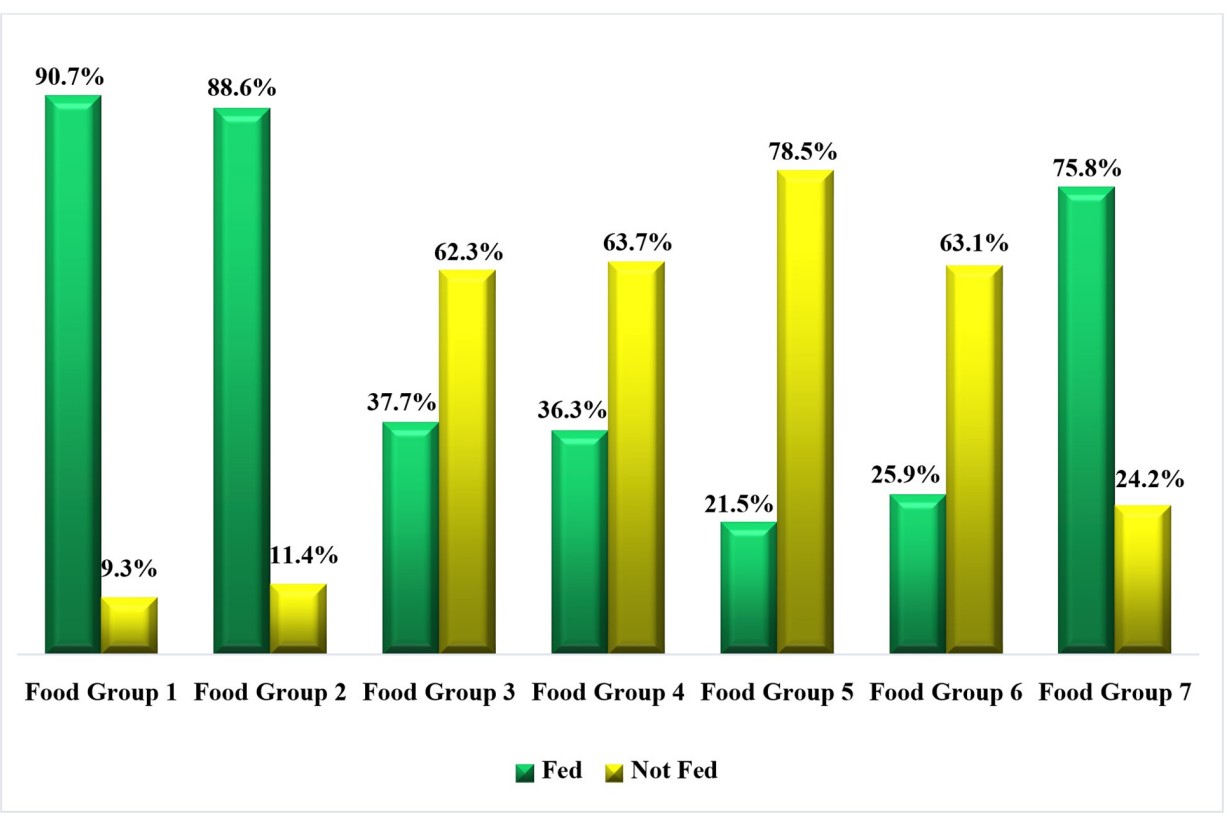

**Fig 4. Proportions distributions of caregivers' feeding practices of dietary diversity based on groups of foods. Source**: Field Data (2022).

### Caregivers' dietary adequacy feeding practices for their under-five children

Findings presented in Fig 5 show that 67.8% (n = 196) of caregivers fed their children inadequate diets. The findings demonstrate that only 32.2% (n = 102) of children were fed adequately. Refer to Fig 5 for a figurative elaboration of the findings.

### Caregivers' dietary adequacy feeding practices to their under-five children based on the WHO recommended Minimum Meal Frequency (MMF)

Findings in Fig 6 display caregivers' feeding practices on the aspect of dietary adequacy based on the WHO-recommended MMF criteria. It was observed that 89.6% (n = 259) of the caregivers fed their children complementary foods as per the WHO-recommended MMF. Moreover, 70.6% (n = 204) of them adhered to the frequency of breastfeeding their children. However, findings show that only 40.9% (n = 118) of the caregivers fed their children snacks between meals based on the recommended frequencies. Refer to Fig 6 for the figurative presentation of the findings.

### Caregivers' factors related to feeding practice

Findings from descriptive analysis through chi-square that established a relationship between caregivers' sociodemographic characteristics profiles and their feeding practices are presented in Table 3. Factors including sex, education level, marital status, the total number of children they have, number of under-five children they have, the place they delivered their babies, their average birth space between children, occupation, and a person who spends much time caring

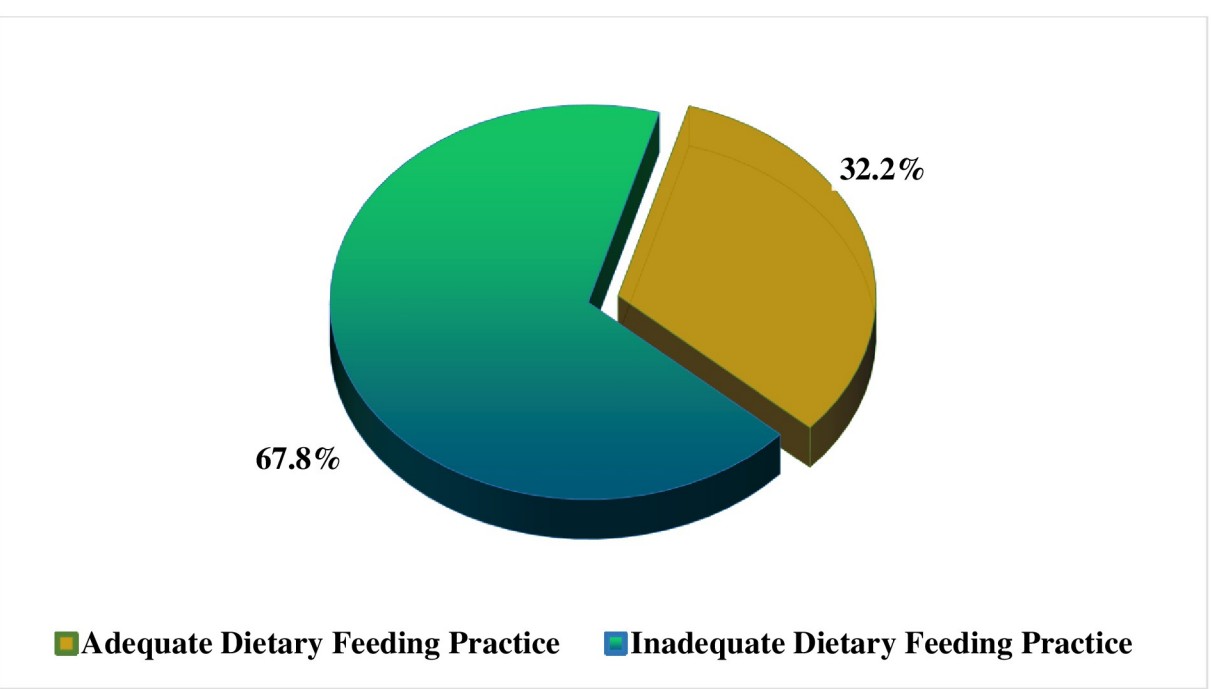

**Fig 5. Caregivers' dietary adequacy feeding practices to their under-five children. Source**: Field Data (2022).

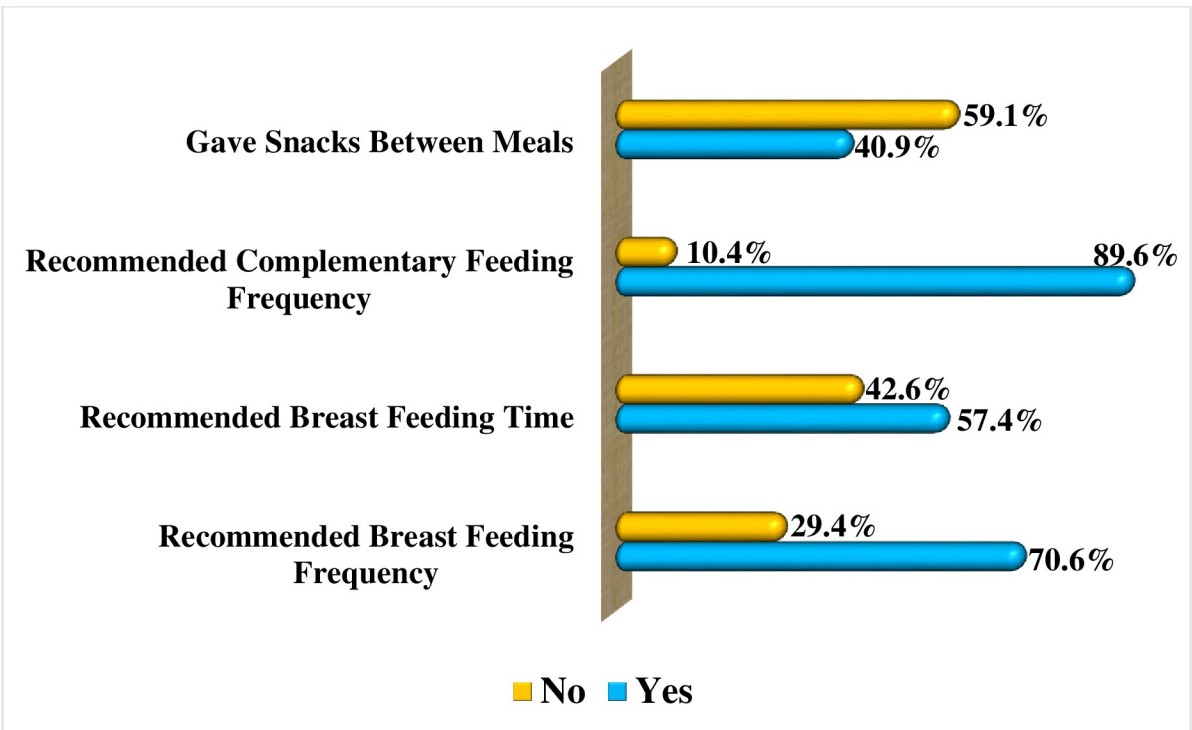

**Fig 6. Caregivers' dietary adequacy feeding practices based on the WHO recommended Minimum Meal Frequency (MMF). Source**: Field Data (2022).

**Table 3. Caregivers' factors related to feeding practice (n = 289).**

| Variable | Feeding practice | | $X^2$(p-value) |
| --- | --- | --- | --- |
| | **Unsatisfactory** | **Satisfactory** | |
| | **n (%)** | **n (%)** | |
| **Age groups** | | | 7.63(0.106) |
| 16–21 yrs. | 54(75.0%) | 18(25.0%) | |
| 22–27 yrs. | 62(58.5%) | 44(41.5%) | |
| 28–33 yrs. | 38(62.3%) | 23(37.7%) | |
| 34–39 yrs. | 32(76.2%) | 10(23.8%) | |
| 40–42 yrs. | 5(62.5%) | 3(37.5%) | |
| **Sex** | | | **14.34(0.001)** |
| Male | 75(81.5%) | 17(18.5%) | |
| Female | 116(58.9%) | 81(41.1%) | |
| **Education Level** | | | **19.86(0.001)** |
| No formal education | 15(100.0%) | 0(0%) | |
| Primary education | 19(100.0%) | 0(0%) | |
| Secondary education | 116(61.1%) | 74(38.9%) | |
| Tertiary education | 41(63.1%) | 24(36.9%) | |
| **Marital Status** | | | **8.89(0.012)** |
| Single | 45(83.3%) | 9(16.7%) | |
| Widow | 13(65.0%) | 7(35.0%) | |
| Married | 133(61.9%) | 82(38.1%) | |
| **Number of children** | | | **22.9(0.001)** |
| Beyond five children | 28(96.6%) | 1(3.4%) | |
| One child | 113(69.8%) | 49(30.2%) | |
| Between two and five children | 50(51.0%) | 48(49.0%) | |
| **Place delivered a baby** | | | **15.54(0.001)** |
| Home | 26(83.9%) | 5(16.1%) | |
| Traditional midwives | 21(95.5%) | 1(4.5%) | |
| Hospital | 144(61.0%) | 92(39.0%) | |
| **Average birth space** | | | **33.68(0.001)** |
| Less than one year average | 49(89.1%) | 6(10.9%) | |
| One year average | 114(69.1%) | 51(30.9%) | |
| Two years average | 28(40.6%) | 41(59.4%) | |
| **Number of under-five children** | | | **10.75(0.001)** |
| Two under-five children | 59(81.9%) | 13(18.1%) | |
| One under-five child | 132(60.8%) | 85(39.2%) | |
| **Occupation** | | | **42.1(0.001)** |
| Don't have a job | 62(95.4%) | 3(4.6%) | |
| Employed | 64(69.6%) | 28(30.4%) | |
| Self-employed | 65(49.2%) | 67(50.8%) | |
| **Proximal childcare person** | | | **25.34(0.001)** |
| Helped by house girl/boy | 48(82.8%) | 10(17.2%) | |
| Helped by grandparents/ relatives | 54(81.8%) | 12(18.2%) | |
| Parent Him/ Herself | 89(53.9%) | 76(46.1%) | |
| **Residency** | | | **4.05(0.044)** |
| Rural | 42(77.8%) | 12(22.2%) | |
| Urban | 149(63.4%) | 86(36.6%) | |
| **Religion** | | | 1.98(0.370) |

*(Continued)*

**Table 3.** (Continued)

| Variable | Feeding practice | | $X^2$(p-value) |
| --- | --- | --- | --- |
| | **Unsatisfactory** | **Satisfactory** | |
| | **n (%)** | **n (%)** | |
| Christian | 81(61.8%) | 50(38.2%) | |
| Muslim | 102(69.4%) | 45(30.6%) | |
| Pagan | 8(72.7%) | 3(27.3%) | |

**Keynotes**: p<0.05 is statistically significant.
**Source**: Field Data (2022).

a baby, residency demonstrated a significant relationship with caregivers' feeding practices respectively (p<0.05). Refer to Table 3 for other findings.

## Caregivers' children factors related to feeding practice

Findings from descriptive analysis through chi-square that established a relationship between children's sociodemographic characteristics profiles and caregivers' feeding practices are presented in Table 4. Factors including sex and nutritional status of children were

**Table 4. Caregivers' children factors related to feeding practice (n = 289).**

| Variable | Feeding practice | | $X^2$(p-value) |
| --- | --- | --- | --- |
| | **Unsatisfactory** | **Satisfactory** | |
| | **n(%)** | **n(%)** | |
| **Children sex** | | | **35.7(0.001)** |
| Boy | 103(85.8%) | 17(14.2%) | |
| Girl | 88(52.1%) | 81(47.9%) | |
| **Children age distribution: Mean age = 2 years±1.2** | | | 8.99(0.061) |
| 1 yrs. | 70(69.3%) | 31(30.7%) | |
| 2 yrs. | 66(73.3%) | 24(26.7%) | |
| 3 yrs. | 26(50.0%) | 26(50.0%) | |
| 4 yrs. | 15(60.0%) | 10(40.0%) | |
| 5 yrs. | 14(66.7%) | 7(33.3%) | |
| **Children's birth weight** | | | 0.192(0.979) |
| Below 3.5 | 20(69.0%) | 9(31.0%) | |
| Above 4.4 | 12(66.7%) | 6(33.3%) | |
| Normal; between 3.5 and 4.4 | 144(65.5%) | 76(34.5%) | |
| Don't remember | 15(68.2%) | 7(31.8%) | |
| **Children vaccination status** | | | 2.91(0.148) |
| Didn't get all appropriate vaccines per age | 43(74.1%) | 15(25.9%) | |
| Got all appropriate vaccines per age | 148(64.1%) | 83(35.9%) | |
| **Children's nutritional status; WHO w/a Z score** | | | **60.25(0.001)** |
| Severely Underweight | 39(97.5%) | 1(2.5%) | |
| Moderately Underweight | 50(94.3%) | 3(5.7%) | |
| Normal Weight | 88(56.4%) | 68(43.6%) | |
| Overweight | 14(35.0%) | 26(65.0%) | |

**Keynotes**: p<0.05 is statistically significant.
**Source**: Field Data (2022).

significantly related to caregivers' feeding practices (p<0.05). Refer to Table 4 for other findings.

## Discussion

The study addressed the issue of feeding practices among caregivers with under-five children in response to the problem of stunting in the Dodoma region, Tanzania. The main focus was to unfold the feeding adequacy and diversification practiced by caregivers to their under-five children at home. Findings revealed that although 32.2% of the caregivers attained feed their under-five children the recommended minimum dietary diversified foods, but, many of them had unsatisfactory feeding practices for their under-five children as most of the children were fed below and some above the world health organization recommended minimum dietary diversity respectively. Unsatisfactory feeding practices were more prominent in the aspect of diversifying foods because the study found that group one, two, and seven foods were the most common foods fed to under-five children than foods from other groups such as group three, four, five, and six.

The observed findings can probably be argued to be the case in this study because caregivers might have no adequate knowledge about the world health organization's recommendations about the types of foods and patterns of feeding under-five children to promote their growth and development. Moreover, the study on hand believes that based on the socio-economic backgrounds of people in Tanzania, it would be possible for caregivers to feed their under-five children groups of foods available, accessible, and affordable to them. The feeding practice situation observed in this study was linked with sociodemographic characteristics profiles of caregivers as most of them were self-employed such that they were busy fighting for food to feed their families. With that in mind, under-five children would be at great risk of not being fed adequately, timely, and with diverse foods based on their age and sex.

Moreover, the situation appeals that spending most of the time fighting for food to feed the family, under-five children would be taken care of by other close relatives such as house girls/ boys, grandmothers, grandfathers, aunts, and/or sisters. Letting individuals distal to a child feed them may imply that children are taken care of by less knowledgeable people on nutrition and therefore they would not get appropriate and recommended groups of foods alongside the recommended frequencies of being fed. Despite the majority of caregivers not being good at feeding practicing, some of them in this study attained to feed their under-five children at the WHO-appropriate and recommended MDD. They did well in some aspects particularly in assuring that their under-five children are fed not only foods from all groups but also the introduction of snacks between meals at the recommended frequencies.

In line with other previous studies [20, 32], the findings of this study demonstrated that a huge proportion of caregivers fed their under-five children complementary feeding and introduced snacks between meals at WHO-appropriate and recommended frequency (least of four times daily). Furthermore, tallying the findings of other scholars [33, 34] has also revealed that caregivers' unsatisfactory feeding practices against the WHO recommendations have been a problem of public concern. The problem is linked significantly with caregivers' sex, education level, marital status, the total number of children they have, the number of under-five children they have, the place they delivered their babies, their average birth space between children, occupation, and a person who spends much time caring a baby, residency [33, 35].

Findings of unsatisfactory feeding practices among caregivers in this study lined up with the work by Belay *et al.,* [12], which exposed that caregivers' characteristics profiles such as education were significantly linked to their habits of feeding their children based on the recommended dietary diversity and frequencies. Moreover, the work by Gebru *et al.,* [10]

cemented that mothers' behaviour, knowledge, attitude, and beliefs may fuel appropriate and recommended feeding practices for under-five children. Nevertheless, findings by Forh *et al.*, [36] insisted that ordinary education of caregivers plus nutrition education, and feeding practices with much attention to dietary diversity and dietary adequacy becomes efficient.

The findings of this study are in support of those from previous scholars who compel that the problem of low feeding practices among caregivers of under-five children persists. Despite any observed mismatch of caregivers' sociodemographic characteristics profiles between studies including differences in geographical locations and methodological approaches, caregivers of Dodoma region, the central part of Tanzania have perceived themselves to have low feeding practices. In support with other previous studies [1, 33], Findings may imply that the situation needs to be addressed accordingly by focusing on community-based educational interventions, which will also put caregivers alongside their sociodemographic characteristics and backgrounds profiles in the first position.

## Conclusion

The findings of this study, caregivers demonstrated unsatisfactory feeding practices thus, under-five children were fed inadequately with very minimal diversification of foods. Under-five children were fed foods under groups one, two, and seven than foods from other groups. The minority of caregivers fed their under-five children snacks between meals per the world health organization MDD and MMF recommendations. Moreover, based on their sex, education levels, and/or occupation, the findings unfolded that some caregivers seemed to entrust house girls/boys or grandmothers/fathers to feed their children in their absence regardless of the truth that they were not having immense nutritional competencies to take such a role. It would be impossible for them to guarantee that house girls/boys, and or grandmothers/fathers could adhere to the recommended MDD and MMF feeding practices in their absence.

Based on the findings of this study, there is still a prominent and unattended problem in feeding practices among caregivers with under-five children at their homes in Dodoma region, the central part of Tanzania. There appears a need to disseminate nutritional-based interventions at the community level to empower caregivers with under-five children at their homes on nutritional competencies per the world health organization MMD and MMF recommendations to protect and promote the growth and development of under-children.

## Recommendations

This study has found that the majority of caregivers with under-five children demonstrate unsatisfactory feeding practices and thus, low dietary adequacy and diversifications. Their sociodemographic characteristics profiles were found to be significantly related to the observed feeding practices. Higher authorities including policymakers, the ministry of health and social welfare, and the ministry of community development, gender, elderly, and children are recommended by this study to revise, strengthen and establish nutritional education programs at easy reach to caregivers at both health facilities and homes.

There may be a need for disseminating nutritional-based interventional services to the community/family levels to promote nutritional competencies among caregivers, house girls, houseboys, and/or grandmothers/fathers through various media including postnatal (hospital facility) locality, social media groups, phone calls, community-based clinics, and/or home visits. Initiatives such as home visiting, phone calls, and/or community-based clinics may be encouraged and given priority to strengthening post-discharge follow-ups of caregivers at their homes to assure appropriate and recommended feeding practices. This will help to

improvise better dietary practices, in terms of adequacy and diversification of foods among the cohort of caregivers with under-five children in Dodoma region, Tanzania.

## The implication of the study for practices

The findings from this study highlight the magnitude of feeding practice problems among caregivers with under-five children who are residing in Dodoma region, Tanzania thus, they can be used in day-to-day hospital-home health activities by health professionals to assure a continuum adherence and implementation of the WHO MDD and MMF recommendation by them in the region. Health organs such as the ministry of health, professional organizations/ unions non-governmental organizations (NGOs), and/or stakeholders in the health field revisit the existing nutritional educational guidelines alongside facilitation pedagogics/andra-gogy and strengthen their scope, implementation, monitoring, and evaluation for the continuum feeding practices among caregivers with under-five children at both health facilities and their homes.

## The implication of the study for future research

If successfully published, the findings of this study may disseminate and may serve as a base for large-scale studies interventions/projects or those that will establish causal-relationship more intensively need to be conducted in the region to address the problem accordingly.

## Strengths of the study

This study has addressed a very important issue in maternal and child health on the aspect of childhood nutrition as advocated by SDG3 target 3.4. Moreover, the study adhered to the research ethics and studied a large sample size at the community level to establish a real-life picture of feeding practices outside health facilities among caregivers of their children.

## Limitations of the study

The study was conducted in a very confined locality and thus findings may not be generalized to caregivers of other geographical locations in or outside the country other than those resid-ing in Dodoma region, the central part of Tanzania. The findings of this study have to be inter-preted carefully since it involved a very small sample size of caregivers and it did not establish causal-relationship between the variables under study. The study moreover, did not use a tri-angulation approach for data collection and thus, the rigor of dependability, transferability, and/or confirmability may have not been addressed in this study. Nevertheless, the findings of this study may need to be interpreted with caution as caregivers would have faced recalling problems remembering and sharing their previous feeding habits. Having an opportunity to rate oneself is criticized as it may influence someone to under or overrate or report the habits/ information as caregivers who participated in the study. Therefore, attention is needed when interpreting the findings of this study.

## Declarations

### Ethics approval and consent to participate

The study adhered to the institution's guidelines and UDOM Institutional Research Review Committee (IRRC) approved it with an approval letter referenced DJ.232/238/0-28. The prin-cipal investigator collected written informed consent from the participating respondents as one of the criteria to join the study.

## Supporting information

**S1 Data.**
(SAV)

## Acknowledgments

It is a privilege to thank our Almighty God for the life and strengths of accomplishing this work. Sincerest gratitude goes to the University of Dodoma (UDOM) and administrative organs of health facilities within Dodoma region for their willingness and support in offering ethical clearances. We acknowledge the willingness and consent of postpartum mothers to join and offer extensive cooperation in providing unlimited information throughout the study. Contributions of the aforementioned organs/people have been substantial to the fruits of this work.

## Author Contributions

**Conceptualization:** Walter C. Millanzi, Patricia Z. Herman, Bariki A. Ambrose.

**Data curation:** Walter C. Millanzi.

**Formal analysis:** Walter C. Millanzi.

**Investigation:** Bariki A. Ambrose.

**Methodology:** Walter C. Millanzi, Patricia Z. Herman, Bariki A. Ambrose.

**Supervision:** Walter C. Millanzi.

**Writing – original draft:** Walter C. Millanzi, Patricia Z. Herman.

**Writing – review & editing:** Walter C. Millanzi, Patricia Z. Herman.

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
