## [Decision Letter · Decision Letter 0]

4 Jan 2023

PONE-D-22-28886Perceived feeding practices, dietary adequacy and dietary diversities among caregivers with under-five children: A descriptive cross-section study in Dodoma region, TanzaniaPLOS ONE

Dear Dr. Millanzi,

Thank you for submitting your manuscript to PLOS ONE. After careful consideration, we feel that it has merit but does not fully meet PLOS ONE’s publication criteria as it currently stands. Therefore, we invite you to submit a revised version of the manuscript that addresses the points raised during the review process.

We look forward to receiving your revised manuscript.

Kind regards,

Charles Odilichukwu R Okpala

Academic Editor

PLOS ONE

Journal Requirements:

“NO - This work did not receive any specific grant from funding agencies in the public, commercial, and or non-profit sectors. It was privately sponsored”

3. Please amend the manuscript submission data (via Edit Submission) to include author “Rabia K. ABDALLAH1”

4.  Please amend your authorship list in your manuscript file to include author Bariki A. Ambrose

Additional Editor Comments:

Reviewers have provided useful comments that will improve this work. Please authors should carefully and diligently revise their work.

Reviewers' comments:

Reviewer's Responses to Questions

**Comments to the Author**

1. Is the manuscript technically sound, and do the data support the conclusions?

Reviewer #1: No

Reviewer #2: Yes

Reviewer #3: Partly

Reviewer #4: Partly

2. Has the statistical analysis been performed appropriately and rigorously? 

Reviewer #1: No

Reviewer #2: Yes

Reviewer #3: Yes

Reviewer #4: Yes

3. Have the authors made all data underlying the findings in their manuscript fully available?

Reviewer #1: Yes

Reviewer #2: Yes

Reviewer #3: Yes

Reviewer #4: No

4. Is the manuscript presented in an intelligible fashion and written in standard English?

Reviewer #1: Yes

Reviewer #2: Yes

Reviewer #3: No

Reviewer #4: No

5. Review Comments to the Author

Reviewer #1: 1. Perception can not be quantified, Qualitative data collection is best to study perception.

2. Wrong order of authors at line 4.

3. No sampling procedure and sample size determination formula indicated.

4. Finding at abstract line 42, 43 contradict that at result in line 229.

5. Result is inadequate due to inefficiency of data collection tools. no evidence for line 320, 321 findings.

6. Recommendation- No new innovative ideas generated by the research.

7. Some findings are misplaced after reference.

Reviewer #2: REVIEWER’S COMMENTS

Perceived feeding practices, dietary adequacy and dietary diversities among caregivers with under-five children: A descriptive cross-section study in Dodoma region, Tanzania

PONE-D-22-28886

GENERAL COMMENTS

This is an interesting study that provides useful insights into perceived feeding practices, dietary adequacy and dietary diversities among caregivers with under-five children. The findings of the research are clearly presented and well-discussed. Comments have been made to improve the final version.

ABSTRACT

1. Line 43: “A 90.7% of children …” is ambiguous. Sentence should be rephrased.

2. Line 47 – 50: Rephrase the first sentence to ensure clarity of message. It is lengthy.

3. Line 52/53: Arrange keywords in alphabetical order.

INTRODUCTION

Line 79: and/or

METHODS AND MATERIALS

Well-written.

RESULTS

1. Line 207, 217: There should be unit for the (mean) age i.e., years

2. Line 209/210, 219: “A 68.2% …” is ambiguous. Sentence is confusing and should be rephrased.

3. Table 1 (2nd Row), Table 2 (5th Row), Table 4 (5th Row): Is that the mean age? If it is, it should be specified.

4. Line 252: Preferably, do not commence sentences with figures written as numbers.

5. Table 3, Table 4: P-values which are statistically significant should be signified.

DISCUSSION

Well-written.

CONCLUSION

The Conclusion should be specific and straightforward. It should provide answers to the main objective of the study. It should be rewritten.

Line 334: Recommendations

Line 345 – 349: Rephrase. Sentence is too long.

Reviewer #3: General Comments: The background created a gap, but the study is a bit complex. First, perception in this study is not well understood. How was perception measured? All I can deduce is feeding practices of caregivers. So, the study should be on “Feeding practices, dietary adequacy and dietary diversities among caregivers with under-five children: A descriptive cross-sectional study in Dodoma region, Tanzania” if the present study would remain as it is. Second, two manuscripts could emerge from this study to make each sharper and simple. For example, one could be on “Feeding practices and dietary diversities among caregivers with under-five children” and the second could be on “dietary adequacy”. I strongly recommend this split.

Methods

• There should be a total population from where the sample was drawn. What is the population of under-five children in the study area? Then, how was the sample size calculated? What were the sampling procedure and techniques used?

• It is not also clear whether the respondents (caregivers) were met in their homes or in common venue. For example, this sentence “Caregivers were reached at their homes and data collection procedures were performed by the principal investigator and trained assistant researchers in an unoccupied venue available at the respective ward to assure privacy” is not clear.

Results

• The results section is cumbersome (four Tables and five figures) and that’s more reason to split the manuscript into two. This will affect the discussion.

Reviewer #4: Thanks for the opportunity to review this paper titled: “Perceived feeding practices, dietary adequacy and dietary diversities among caregivers with under-five children: A descriptive cross-section study in Dodoma region, Tanzania” This addresses an important issue in public health, however, the quality of the paper is marred by so many issues (including grammatical errors) which the authors need to address. For instance, there are so many verb agreement and punctuation mistakes throughout the manuscript, and I have tried to indicate some of them. Also, the paper requires major English editing. Therefore, the authors should subscribe to an editing service for adequate and professional editing of this paper.

Abstract

• The abstract should be presented in a clearer manner with more findings included in the result subsection.

• Line 29: ‘Feeding newborn while at health facilities assure…’(there is a verb agreement mistake in this clause)

• Line 35: (not clear)

• Line 44: ‘A 90.7% of children were fed group one 44 foods of which 59.1% (n =152) of them were not fed …’(there is an agreement mistakes in this

• Line 66: ‘Proper adherence to dietary adequacy, improvising dietary diversities and proper feeding practices is be’ (verb agreement mistake)

Introduction

• This needs better organisation for smooth flow. The messages are scattered all over the section. It’s good to start with global situation, then African/LMIC and finally Tanzania. Each paragraph should have just one message

• Line 68: Existing 68 knowledge uncover that (verb agreement mistake)

• The authors should make a stronger justification for the paper and show how the paper contributes to new knowledge.

Methodology

• Relevant characteristics of the study area/site are missing in this manuscript

• Considering the sampling technique, more details, such as the sampling interval and sampling ratio should be provided.

• How was the sample size calculated, and which software program was used?

• Write in details about the study participants, selection criteria (inclusion and exclusion).

• Were questionnaire structured?

Result

• The abstract and the methodology indicate that 298 caregivers were sampled and the response rate was 100%, why did the authors report only 289 in the result section?

• The interpretations of the results should be better stated for clarity

Discussion

• The discussion is weakly presented and did not address most of the findings in the result section.

• There is a mix up of different referencing styles (lines 301-310)

• The authors may consider starting the discussion with a summary statement that showcases the overall message of the study

• Line 358: apart from addressing a very important issue in maternal and child health, what are the strengths of this study?

Conclusion

This should be revised and better presented to capture the key findings of the study

• 320: ‘Even though many caregivers are empowered with nutritional knowledge during their reproductive health clinic (RCH) visits and various scholarly…’ this does not appear to be part of the finding of this study

Ethical Approval:

The number of the ethical approval from the Health Research Ethics Committee of the institution should be indicated.

6. PLOS authors have the option to publish the peer review history of their article (what does this mean?). If published, this will include your full peer review and any attached files.

Reviewer #1: No

Reviewer #2: **Yes: **Kosisochi Chinwendu Amorha

Reviewer #3: **Yes: **Amelia Ngozi Odo

Reviewer #4: No

---

## [Author Response · Author response to Decision Letter 0]

20 Jan 2023

Feeding practices, dietary adequacy and dietary diversities among caregivers with under-five children: A descriptive cross-section study in Dodoma region, Tanzania

Ref: Submission ID: PONE-D-22-28886

General: Thank you for submitting your manuscript to PLOS ONE. After careful consideration, we feel that it has merit but does not fully meet PLOS ONE’s publication criteria as it currently stands. Therefore, we invite you to submit a revised version of the manuscript that addresses the points raised during the review process

JOURNAL REQUIREMENTS 

S/N REVIEWER’S COMMENT AUTHOR’S RESPONSE PG. NUMBER

1 1. Please ensure that your manuscript meets PLOS ONE's style requirements, including those for file naming. The PLOS ONE style templates can be found at

The manuscript has been revised to meet the PLO ONE’s style requirements Pg. 1-18 of the revised Manuscript

 Thank you for stating the following financial disclosure:

“NO - This work did not receive any specific grant from funding agencies in the public, commercial, and or non-profit sectors. It was privately sponsored”

a) Please clarify the sources of funding (financial or material support) for your study. List the grants or organizations that supported your study, including funding received from your institution. The funding section has been revised and the financial support information has be made more clearer that the work did not receive any financial or materials support from any financial agencies/organization Pg. 14 of the revised Manuscript

&

Journal/author’s manuscript submission system

 b) State what role the funders took in the study. If the funders had no role in your study, please state: “The funders had no role in study design, data collection and analysis, decision to publish, or preparation of the manuscript.” The funding section has been revised and made more clearer that “Since no funder was involved in this study, no role concerning them has been provided” Pg. 14 of the revised Manuscript

&

Journal/author’s manuscript submission system

 c) If any authors received a salary from any of your funders, please state which authors and which funders. The funding section has been revised and made more clearer that “No any authors received a salary from any of your funders, please state which authors and which funders” Pg. 14 of the revised Manuscript

&

Journal/author’s manuscript submission system

Please include your amended statements within your cover letter; we will change the online submission form on your behalf The funding section has been revised and made clearer that ““The authors received no specific funding for this work.” Pg. 14 of the revised Manuscript

&

Journal/author’s manuscript submission system

2 Please amend the manuscript submission data (via Edit Submission) to include author “Rabia K. ABDALLAH1” The name “Rabia K. Abdallah” has been revised and removed as she’s not part of this work and thus, authors declare that it was a typing error. Pg. 1 (Title page) of the revised Manuscript

&

Journal/author’s manuscript submission system

 Please amend your authorship list in your manuscript file to include author Bariki A. Ambrose The name “Rabia K. Abdallah” has been revised and removed as she’s not part of this work and thus, authors declare that it was a typing error. Pg. 1 (Title page) of the revised Manuscript

&

Journal/author’s manuscript submission system

3 Please include your full ethics statement in the ‘Methods’ section of your manuscript file. In your statement, please include the full name of the IRB or ethics committee who approved or waived your study, as well as whether or not you obtained informed written or verbal consent. If consent was waived for your study, please include this information in your statement as well. The research ethics part in the methods section has been revised and the name of the IRRC mentioned, informed consent statement added accordingly Pg. 14 of the revised Manuscript

REVIEWER #1: COMMENTS TO THE AUTHOR

S/N REVIEWER’S COMMENT AUTHOR’S RESPONSE PG. NUMBER

1 Perception cannot be quantified, Qualitative data collection is best to study perception The word “perceived” has been revised and removed because the study did not assess participants’ perception of feeding but, their practices on it. Authors’ declare that the word was misused Pg. 1 – 14 of the revised Manuscript

2 Wrong order of authors at line 4 The order of authors has been revised and re-ordered accordingly Pg. 14 of the revised Manuscript

&

Journal/author’s manuscript submission system

3 No sampling procedure and sample size determination formula indicated Sample size and sampling procedures have been revised and added/indicated accordingly Pg. 7 & 8 of the revised Manuscript 

4 Finding at abstract line 42, 43 contradict that at result in line 229 Findings in the abstract have been revised so that they tally with those in the result section Pg. 2 of the revised Manuscript

5 Result is inadequate due to inefficiency of data collection tools. No evidence for line 320, 321 findings Authors acknowledge the comment. The methods and results section has been revised and improved accordingly. However, findings presented in this work addressed the variables under study only because findings of this study serve as descriptive baseline observations to the planned intervention and thus, they may be seen inadequate Pg. 7-22 of the revised Manuscript

6 Recommendation- No new innovative ideas generated by the research The recommendation section has been revised and re-written to reflect and demonstrate what has been found in the current study based on the variables under study Pg. 23 of the revised Manuscript

7 Some findings are misplaced after reference The findings and the reference sections have been revised and improved accordingly so that no findings are misplaced after reference Pg. 12-32 of the revised Manuscript

REVIEWER #2: COMMENTS TO THE AUTHOR

General comments

This is an interesting study that provides useful insights into perceived feeding practices, dietary adequacy and dietary diversities among caregivers with under-five children. The findings of the research are clearly presented and well-discussed. Comments have been made to improve the final version.

S/N REVIEWER’S COMMENT AUTHOR’S RESPONSE PG. NUMBER

Abstract Line 43: “A 90.7% of children …” is ambiguous. Sentence should be rephrased

 Line 43 has been revised and improved accordingly Pg. 2 of the revised Manuscript

 Line 47 – 50: Rephrase the first sentence to ensure clarity of message. It is lengthy Line 47-50 have been revised and improved accordingly to ensure clarity as suggested Pg. 3 of the revised Manuscript

 Line 52/53: Arrange keywords in alphabetical order Line 52/53 have been revised and the keywords arranged alphabetically Pg. 3 of the revised Manuscript 

Introduction Line 79: and/or Line 79 has been revised and improved accordingly Pg. 4 of the revised Manuscript

Methods Well-written Authors acknowledges the reviewer’s comment about the methods and materials section Pg. 6-12 of the revised Manuscript

Results Line 207, 217: There should be unit for the (mean) age i.e., years Line 207, 2017 have been revised and the unit for age written accordingly Pg. 12 of the revised Manuscript

 Line 209/210, 219: “A 68.2% …” is ambiguous. Sentence is confusing and should be rephrased Line 209/210, 2019 have been revised and the sentence re-written accordingly Pg. 12 of the revised Manuscript

 Table 1 (2nd Row), Table 2 (5th Row), Table 4 (5th Row): Is that the mean age? If it is, it should be specified Table 1 (2nd Row), Table 2 (5th Row), Table 4 (5th Row) have been revised and specified that in they are mean ages respectively Pg. 13, 14, & 19 of the revised Manuscript

 Line 252: Preferably, do not commence sentences with figures written as numbers

 Line 252 has been revised and re-written accordinlgy Pg. 12 - 19 of the revised Manuscript

 Table 3, Table 4: P-values which are statistically significant should be signified p-values in Table 3 and 4 have been revised and signified

 Pg. 17-19 of the revised Manuscript

Discussion Well-written Authors acknowledges the reviewer’s comment about the methods and materials section Pg. 19-22 of the revised Manuscript

Conclusion The Conclusion should be specific and straightforward. It should provide answers to the main objective of the study. It should be rewritten The conclusion section has been revised and re-written accordingly Pg.22 of the revised Manuscript

 Line 334: Recommendations The recommendation section has been revised and re-written accordingly Pg.23 of the revised Manuscript

 Line 345 – 349: Rephrase. Sentence is too long Line 345-349 have been revised and rephrased accordingly Pg. 23 of the revised Manuscript

REVIEWER #3: COMMENTS TO THE AUTHOR

General Comments: The background created a gap, but the study is a bit complex. First, perception in this study is not well understood. How was perception measured? All I can deduce is feeding practices of caregivers. So, the study should be on “Feeding practices, dietary adequacy and dietary diversities among caregivers with under-five children: A descriptive cross-sectional study in Dodoma region, Tanzania” if the present study would remain as it is. Second, two manuscripts could emerge from this study to make each sharper and simple. For example, one could be on “Feeding practices and dietary diversities among caregivers with under-five children” and the second could be on “dietary adequacy”. I strongly recommend this split.

S/N REVIEWER’S COMMENT AUTHOR’S RESPONSE PG. NUMBER

General comments First, perception in this study is not well understood. How was perception measured? The word “perceived” has been revised and removed because the study did not assess participants’ perception of feeding but, their practices on it. Authors’ declare that the word was misused Pg.1- 23 of the revised Manuscript

 So, the study should be on “Feeding practices, dietary adequacy and dietary diversities among caregivers with under-five children: A descriptive cross-sectional study in Dodoma region, Tanzania” if the present study would remain as it is The study title has been revised and the word “perceived/perception” omitted as suggested Pg. 1 of the revised Manuscript

 Two manuscripts could emerge from this study to make each sharper and simple. For example, one could be on “Feeding practices and dietary diversities among caregivers with under-five children” and the second could be on “dietary adequacy”. I strongly recommend this split Authors have acknowledged the reviewers recommendation on the split of two manuscripts from this study. However, based on the ethical clearances of the institution and being the baseline findings for the coming interventional study and avoid the repetition of the concepts/context we would advise it remain in its current state as it is Pg. 1-32 of the revised Manuscript 

Methods There should be a total population from where the sample was drawn:

What is the population of under-five children in the study area? 

Then, how was the sample size calculated? 

What were the sampling procedure and techniques used? Sample size and sampling procedures have been revised and the details about them added/indicated accordingly Pg. 7-8 of the revised Manuscript

 It is not also clear whether the respondents (caregivers) were met in their homes or in common venue. For example, this sentence “Caregivers were reached at their homes and data collection procedures were performed by the principal investigator and trained assistant researchers in an unoccupied venue available at the respective ward to assure privacy” is not clear The data collection procedures section has been revised and re-written to improve clarity as suggested Pg. 10 of the revised Manuscript

Results The results section is cumbersome (four Tables and five figures) and that’s more reason to split the manuscript into two. This will affect the discussion The result and discussion sections have been revised so that they tally in a simple and clear way. Dietary adequacy and diversifications have been treated in this study as the patterns/characterization of feeding practices. The discussion has also revised to demonstrate the key findings on feeding practices alongside its characterization and determinants Pg. 12-22 of the revised Manuscript

REVIEWER #4: COMMENTS TO THE AUTHOR

Thanks for the opportunity to review this paper titled: “Perceived feeding practices, dietary adequacy and dietary diversities among caregivers with under-five children: A descriptive cross-section study in Dodoma region, Tanzania” This addresses an important issue in public health, however, the quality of the paper is marred by so many issues (including grammatical errors) which the authors need to address. For instance, there are so many verb agreement and punctuation mistakes throughout the manuscript, and I have tried to indicate some of them. Also, the paper requires major English editing. Therefore, the authors should subscribe to an editing service for adequate and professional editing of this paper

S/N REVIEWER’S COMMENT AUTHOR’S RESPONSE PG. NUMBER

General comments The quality of the paper is marred by so many issues (including grammatical errors) which the authors need to address. For instance, there are so many verb agreement and punctuation mistakes throughout the manuscript, and I have tried to indicate some of them. Also, the paper requires major English editing. Therefore, the authors should subscribe to an editing service for adequate and professional editing of this paper The manuscript has been revised and improved for the grammatical errors and English editing Pg. 1-32 of the revised Manuscript

Abstract The abstract should be presented in a clearer manner with more findings included in the result subsection The abstract has been revised and the result section improved accordingly Pg. 2 of the revised Manuscript

 Line 29: ‘Feeding newborn while at health facilities assure…’(there is a verb agreement mistake in this clause) Line 29 has been revised and improved accordingly to make it clearer Pg. 2 of the revised Manuscript 

 Line 35: (not clear) Line 35 has been revised and improved accordingly Pg. 2 of the revised Manuscript

 Line 44: ‘A 90.7% of children were fed group one 44 foods of which 59.1% (n =152) of them were not fed …’(there is an agreement mistakes in this) Line 44 has been revised and re-written accordingly to make it clearer Pg. 2 of the revised Manuscript

 Line 66: ‘Proper adherence to dietary adequacy, improvising dietary diversities and proper feeding practices is be’ (verb agreement mistake) Line 66 has been revised and re-written to make it clearer Pg. 4 of the revised Manuscript

Introduction This needs better organization for smooth flow. The messages are scattered all over the section. It’s good to start with global situation, then African/LMIC and finally Tanzania. Each paragraph should have just one message The introduction section has been revised and re-written to establish a clear flow of the concepts for easy understanding Pg. 4-6 of the revised Manuscript

 Line 68: Existing 68 knowledge uncover that (verb agreement mistake) Line 68 has been revised and re-written accordingly to make it clearer Pg. 4 of the revised Manuscript

 The authors should make a stronger justification for the paper and show how the paper contributes to new knowledge Authors have revised the manuscript and try to demonstrate the contribution of the paper that it tries to reveal the unknown about home feeding practices among caregivers Pg. 1-24 of the revised Manuscript

Methods Relevant characteristics of the study area/site are missing in this manuscript The methods part in the study setting section has been revised and improved accordingly

 Pg. 7 of the revised Manuscript

 Considering the sampling technique, more details, such as the sampling interval and sampling ratio should be provided Sampling procedures section has been revised and more details added to make it clearer Pg. 7-8 of the revised Manuscript

 How was the sample size calculated, and which software program was used? Sample size determination section has been revised and more details added to make it clearer Pg.7-8 of the revised Manuscript

 Write in details about the study participants, selection criteria (inclusion and exclusion) Participants’ selection criteria has been revised and the section added accordingly Pg. 8 - 9 of the revised Manuscript

 Were questionnaire structured? The data collection tools section has been revised and questionnaires have been specified based on the nature of items and administration Pg. 9 of the revised Manuscript

Results The abstract and the methodology indicate that 298 caregivers were sampled and the response rate was 100%, why did the authors report only 289 in the result section? Authors acknowledge the discrepancy in the sample size between sections. This was the typing error. Sample size in the result section is correct and thus, changes in the abstract and methods sections made accordingly Pg. 2, & 8 of the revised Manuscript

 The interpretations of the results should be better stated for clarity The results section has been revised and improved accordingly as suggested Pg. 12-20 of the revised Manuscript 

Discussion The discussion is weakly presented and did not address most of the findings in the result section The discussion section has been revised and re-written to reflect the study findings as suggested Pg. 20-22 of the revised Manuscript

 There is a mix up of different referencing styles (lines 301-310) The citation and references have been revised and improved accordingly Pg. 20-22 of the revised Manuscript

 The authors may consider starting the discussion with a summary statement that showcases the overall message of the study The discussion section has been revised and improved accordingly to show the overall message of the study as suggested Pg. 20-22 of the revised Manuscript

 Line 358: apart from addressing a very important issue in maternal and child health, what are the strengths of this study? The strengths section of the study on hand have been revised and more details added accordingly as suggested Pg. 25 of the revised Manuscript

Conclusion This should be revised and better presented to capture the key findings of the study The conclusion section has been revised and improved accordingly to reflect the study findings Pg. 22-23 of the revised Manuscript

 320: ‘Even though many caregivers are empowered with nutritional knowledge during their reproductive health clinic (RCH) visits and various scholarly…’ this does not appear to be part of the finding of this study Line 320 has been revised and modified accordingly to reflect the study findings Pg.22-23 of the revised Manuscript

Ethical approval The number of the ethical approval from the Health Research Ethics Committee of the institution should be indicated The research ethics section has been revised and improved accordingly by a indicating the number of ethical approval from the IRRC Pg. 25-26 of the revised Manuscript

---

## [Decision Letter · Decision Letter 1]

24 Feb 2023

PONE-D-22-28886R1Feeding practices, dietary adequacy and dietary diversities among caregivers with under-five children: A descriptive cross-section study in Dodoma region, TanzaniaPLOS ONE

Dear Dr. Millanzi,

Thank you for submitting your manuscript to PLOS ONE. After careful consideration, we feel that it has merit but does not fully meet PLOS ONE’s publication criteria as it currently stands. Therefore, we invite you to submit a revised version of the manuscript that addresses the points raised during the review process.

ACADEMIC EDITOR: Minor revision required before it can be accepted for publication.

We look forward to receiving your revised manuscript.

Kind regards,

Charles Odilichukwu R Okpala

Academic Editor

PLOS ONE

Journal Requirements:

Additional Editor Comments:

The reviewers have attended to the revised manuscript. Please, kindly attend to the minor concerns remaining. Thank you.

Reviewers' comments:

Reviewer's Responses to Questions

**Comments to the Author**

1. If the authors have adequately addressed your comments raised in a previous round of review and you feel that this manuscript is now acceptable for publication, you may indicate that here to bypass the “Comments to the Author” section, enter your conflict of interest statement in the “Confidential to Editor” section, and submit your "Accept" recommendation.

Reviewer #2: (No Response)

Reviewer #4: (No Response)

2. Is the manuscript technically sound, and do the data support the conclusions?

Reviewer #2: Yes

Reviewer #4: Yes

3. Has the statistical analysis been performed appropriately and rigorously? 

Reviewer #2: Yes

Reviewer #4: Yes

4. Have the authors made all data underlying the findings in their manuscript fully available?

Reviewer #2: Yes

Reviewer #4: Yes

5. Is the manuscript presented in an intelligible fashion and written in standard English?

Reviewer #2: Yes

Reviewer #4: Yes

6. Review Comments to the Author

Reviewer #2: GENERAL COMMENTS

The manuscript should be accepted for publication after the few corrections, stated below, have been addressed.

ABSTRACT

Results: Preferably, do not start sentences with figures written as numbers. Correct this, wherever applicable, in the manuscript.

INTRODUCTION

Lines 80, 83, 408: Include a slash “and/or”

RESULTS

1. Line 252: Preferably, do not commence sentences with figures written as numbers.

2. Table 3, Table 4: keynotes should include “P < 0.05 is statistically significant”

Line 419: “Recommendations” not Recommendation

Line 488: “Acknowledgements” not “Acknowledgement.

Reviewer #4: The authors have tried to improve the general quality of this manuscript in the revised version, and have attended to most of the issues I raised, following my first review of the paper. I commend them. However, I still have a couple of concerns that I would like addressed.

Study setting

This paper focuses on “dietary practices, adequacy and diversities…” among caregivers.

The high prevalence of stunting in the area was stated, but the authors failed to include such details as climatic conditions/vegetation in relation to food security and the socio-cultural beliefs related to infant and young child feeding, in the description of study setting.

“The region was sampled owing to the growing prevalence…”

How was it sampled? (Purposively? randomly? Etc.)…this should be under sampling technique

Study population

Inclusion criteria: “… a resident of Dodoma region for at least more than six months,”

What is your reason for choosing residency period of at least more than six months?

Exclusion criteria: Deaf, dumb, mentally unsound, reported sicknesses

What is the rationale for excluding these groups of people, especially the deaf and dumb? I think excluding the deaf and dumb among the study group raises some ethical questions of fairness. Kindly explain in details why deaf and dumb made the exclusion criteria. How will you establish those who are mentally unsound?

7. PLOS authors have the option to publish the peer review history of their article (what does this mean?). If published, this will include your full peer review and any attached files.

Reviewer #2: **Yes: **Dr. Kosisochi Chinwendu Amorha

Reviewer #4: No

---

## [Author Response · Author response to Decision Letter 1]

24 Feb 2023

Feeding practices, dietary adequacy and dietary diversities among caregivers with under-five children: A descriptive cross-section study in Dodoma region, Tanzania

Ref: Submission ID: PONE-D-22-28886R1

REVIEWER #2 

General: The manuscript should be accepted for publication after the few corrections, stated below, have been addressed.

S/N REVIEWER’S COMMENT AUTHOR’S RESPONSE PG. NUMBER

Abstract Results: Preferably, do not start sentences with figures written as numbers. Correct this, wherever applicable, in the manuscript. The result part of the abstract and the rest of the manuscript has been revised not to start sentences with figures/numbers as suggested Pg. 2-25 of the revised Manuscript

Introduction Lines 80, 83, 408: Include a slash “and/or” Line 80, 83, 408 and the rest of the work has been revised and a slash “and/or” has been included Pg. 2-25 of the revised Manuscript

Results Line 252: Preferably, do not commence sentences with figures written as numbers. Line 252 have been revised not to start sentences with figures written as numbers as suggested Pg. 12 of the revised Manuscript

 Table 3, Table 4: keynotes should include “P < 0.05 is statistically significant” Table 3 and Table 4 have been revised and the key notes have included “p<0.05 is statistically significant as suggested Pg. 18-20 of the revised Manuscript

 Line 419: “Recommendations” not Recommendation Line 419 has been revised and the word “Recommendation” added “s” at the end to be “Recommendations” Pg. 23 of the revised Manuscript

 Line 488: “Acknowledgements” not “Acknowledgement. Line 488 has been revised and the word “Acknowledgement” has been added “s” to be “Acknowledgements” Pg. 27 of the revised Manuscript

REVIEWER #4

The authors have tried to improve the general quality of this manuscript in the revised version, and have attended to most of the issues I raised, following my first review of the paper. I commend them. However, I still have a couple of concerns that I would like addressed

S/N REVIEWER’S COMMENT AUTHOR’S RESPONSE PG. NUMBER

Methods Study setting

-This paper focuses on “dietary practices, adequacy and diversities…” among caregivers.

-The high prevalence of stunting in the area was stated, but the authors failed to include such details as climatic conditions/vegetation in relation to food security and the socio-cultural beliefs related to infant and young child feeding, in the description of study setting 

The study setting section has been revised and improved accordingly to include and link the climatic condition to food security and socio-cultural beliefs to feeding practices

Pg. 7 of the revised Manuscript

 “The region was sampled owing to the growing prevalence…”

-How was it sampled? (Purposively? randomly? Etc.)…this should be under sampling technique Sampling procedures section has been revised and more details added to make it clearer on how the study setting was sampled Pg. 8-9 of the revised Manuscript

 Study population:

Inclusion criteria: “… a resident of Dodoma region for at least more than six months,”

-What is your reason for choosing residency period of at least more than six months? Inclusion criteria of the study population has been revised and the period limit has been removed as it had no any significant rationale Pg.7-8 of the revised Manuscript

 Exclusion criteria: Deaf, dumb, mentally unsound, reported sicknesses.

-What is the rationale for excluding these groups of people, especially the deaf and dumb? I think excluding the deaf and dumb among the study group raises some ethical questions of fairness.

-Kindly explain in details why deaf and dumb made the exclusion criteria. How will you establish those who are mentally unsound? Participants’ exclusion criteria has been revised and improved accordingly based on ethical issues Pg. 9 of the revised Manuscript

---

## [Editor Report · Decision Letter 2]

1 Mar 2023

Feeding practices, dietary adequacy and dietary diversities among caregivers with under-five children: A descriptive cross-section study in Dodoma region, Tanzania

PONE-D-22-28886R2

Dear Dr. Millanzi,

We’re pleased to inform you that your manuscript has been judged scientifically suitable for publication and will be formally accepted for publication once it meets all outstanding technical requirements.

Kind regards,

Charles Odilichukwu R Okpala

Academic Editor

PLOS ONE

Additional Editor Comments (optional):

Thank you authors for revising your work. It is now acceptable for publication.
---

## [Editor Report · Acceptance letter]

14 Mar 2023

PONE-D-22-28886R2 

Feeding practices, dietary adequacy, and dietary diversities among caregivers with under-five children: A descriptive cross-section study in Dodoma region, Tanzania 

Dear Dr. Millanzi:

I'm pleased to inform you that your manuscript has been deemed suitable for publication in PLOS ONE. Congratulations! Your manuscript is now with our production department. 

Kind regards, 

on behalf of

Dr. Charles Odilichukwu R. Okpala 

Academic Editor

PLOS ONE